# Duchenne muscular dystrophy in Italy: A systematic review of epidemiology, quality of life, treatment adherence, and economic impact

**Massimiliano Orso**[1]*, **Antonio Migliore**[1], **Barbara Polistena**[1,2], **Eleonora Russo**[3], **Francesca Gatto**[3], **Mauro Monterubbianesi**[3], **Daniela d'Angela**[1,2], **Federico Spandonaro**[1,4], **Marika Pane**[5,6]

1 C.R.E.A. Sanità (Centre for Applied Economic Research in Healthcare), Rome, Italy, 2 University of Rome Tor Vergata, Rome, Italy, 3 Medical Department, Pfizer Italia, Rome, Italy, 4 San Raffaele University, Rome, Italy, 5 Centro Clinico Nemo, Fondazione Policlinico Universitario Agostino Gemelli IRCCS, Rome, Italy, 6 Pediatric Neurology, Università Cattolica del Sacro Cuore, Rome, Italy

* massi.orso@hotmail.it

**Data Availability Statement:** All relevant data are within the paper and its Supporting Information files.

## Abstract

### Objective

This systematic review aims to update the evidence on Duchenne muscular dystrophy (DMD) in Italy, describing the epidemiology, quality of life (QoL) of patients and caregivers, treatment adherence, and economic impact of DMD.

### Methods

Systematic searches were conducted in PubMed, Embase and Web of Science up to January 2023. Literature selection process, data extraction and quality assessment were performed by two independent reviewers. Study protocol was registered in PROSPERO (CRD42021245196).

### Results

Thirteen studies were included. The prevalence of DMD in the general population is 1.7–3.4 cases per 100,000, while the birth prevalence is 21.7–28.2 per 100,000 live male births. The QoL of DMD patients and caregivers is lower than that of healthy subjects, and the burden for caregivers of DMD children is higher than that of caregivers of children with other neuromuscular disorders. The compliance of real-world DMD care to clinical guidelines recommendations in Italy is lower than in other European countries. The annual cost of illness for DMD in Italy is € 35,000–46,000 per capita while, adding intangible costs, the total cost amounts to € 70,000.

### Conclusion

Although it is a rare disease, DMD represents a significant burden in terms of quality of life of patients and their caregivers, and economic impact.

**Funding:** Writing and editorial support for this review was provided by Massimiliano Orso, Antonio Migliore, Barbara Polistena, Daniela d'Angela, Federico Spandonaro at C.R.E.A. Sanità and by Marika Pane and was funded by Pfizer.

**Competing interests:** I have read the journal's policy and the authors of this manuscript have the following competing interests: Barbara Polistena declares to have received in the last 5 years payments or honoraria for lectures, presentations, speakers bureaus, manuscript writing or educational events from the following commercial sources: Allergan, Amgen, Astellas, BMS, Boehringer-Ingelheim, Celgene, Eli Lilly, Janssen Cilag, Nestle´ HS, Novartis, Novo Nordisk, Pfizer, Roche, Sanofi, Servier, Shire, Takeda, Teva; in addition, she received consulting fees from UCB. Federico Spandonaro declares to have received in the last 5 years payments or honoraria for lectures, presentations, speakers bureaus, manuscript writing or educational events from the following commercial sources: Allergan, Amgen, Astellas, Baxter, BMS, Boehringer-Ingelheim, Celgene, Eli Lilly, Janssen Cilag, Jazzpharma, Mylan, Nestle´ HS, Novartis, Novo Nordisk, Pfizer, Roche, Sanofi, Servier, Shire, Takeda, Teva; in addition, he received consulting fees from Amgen. Marika Pane declares to have received consulting fees for this paper from Pfizer. Eleonora Russo, Francesca Gatto, and Mauro Monterubbianesi are employees of Pfizer. All other authors declare that they have no competing interests.

## 1. Introduction

Duchenne muscular dystrophy (DMD) is a rare X-linked recessive neuromuscular disorder caused by mutations in the dystrophin gene that lead to absent or insufficient functional dystrophin, a cytoskeletal protein that enables the strength, stability, and functionality of myofibres [1]. DMD predominantly affects boys, with the majority of females being asymptomatic carriers [2].

A recent systematic review estimated a global DMD prevalence of 7.1 cases per 100,000 males, and a global birth prevalence of 19.8 per 100,000 live male births [3].

DMD has a severe prognosis, and causes a progressive muscular damage and degeneration, resulting in muscular weakness, associated motor delays, loss of ambulation, respiratory impairment, and cardiomyopathy [1]. Worldwide, most patients are diagnosed at around 5 years of age, when their physical abilities diverge considerably from those of their peers if muscle strength deteriorates and boys require the use of a wheelchair before adolescence [4]. An Italian study on 384 boys diagnosed with DMD from 2005 to 2014 reported that the mean age at diagnosis was 41 months [5]. Median life expectancy in DMD patients not receiving ventilatory support ranges from 14.4 to 27.0 years, while in patients with a ventilatory support it is notably higher, ranging between 21.0 and 39.6 years [6].

DMD requires a multidisciplinary management, involving neurologists or paediatric neurologists, rehabilitation specialists, neurogeneticists, paediatricians, and primary-care physicians [4]. Physiotherapy and treatment with glucocorticoids remain the main therapeutic options for DMD and should continue after the loss of ambulation [1]. Other treatment strategies include surgery for the correction of scoliosis, assisted cough and ventilation support, use of ankle-foot orthoses, cardiac management, and psychosocial care [7, 8]. To optimize the treatment results, it is important that patients adhere to care recommendations; however, some studies showed a suboptimal compliance [9–12].

Patients with DMD usually live at home with their parents for their entire life. As a result, patients with DMD receive most of their daily long-term care from informal family caregivers. This includes emotional and social support and assistance with activities of daily life (e.g., transfers, meal preparation, cleaning, dressing, eating, and toileting), as well as help with administration, organization and the consumption of formal health care [13]. Therefore, in addition to having an impact on the quality of life of patients, DMD also affects that of the caregivers.

Another important aspect to consider is the economic impact of DMD. Over the past two decades, improved standards of care have led to a significant increase in the life expectancy of patients with DMD [6]; at the same time, the increase in the duration of therapies has generated an increase in the economic impact of the disease, both from the perspective of the society and of the patient/household.

This systematic review aims to estimate the burden of DMD in Italy, investigating its epidemiology, the quality of life of DMD patients and their caregivers, the adherence to different treatment/management strategies, and the economic impact of disease.

## 2. Methods

The protocol of this systematic review is registered in PROSPERO (CRD42021245196). The review was conducted and reported in accordance with the Preferred Reporting Items for Systematic Reviews and Meta-Analyses (PRISMA) statement [14]. The PRISMA reporting checklist is available as supplementary material (S1 Appendix).

### 2.1. Review questions

The research questions addressed in this systematic review are:

1. the prevalence and the incidence of DMD in Italy;

2. the quality of life of patients with DMD and their caregivers in Italy;

3. the adherence to the different management strategies for DMD in Italy, such as pharmaco-therapy, physiotherapy and occupational therapy, and to identify factors that could facilitate or hamper the adherence;

4. the economic impact of DMD in Italy.

## 2.2. Literature searches

Systematic literature searches were performed in PubMed, Embase and Web of Science to retrieve studies published from January 2010 to January 2023. Only articles written in English or Italian were considered. The full search strategies are described in S2 Appendix.

## 2.3. Inclusion criteria

- Research question 1 (epidemiology): we included both secondary studies (systematic or narrative reviews) and primary studies (cross-sectional, cohort) describing prevalence and/or incidence of DMD in Italy;

- Research question 2 (quality of life): studies of any design assessing the quality of life of DMD patients and their caregivers, both comparative (e.g. DMD vs healthy subjects) or not;

- Research question 3 (adherence): studies of any design assessing the adherence to the different management strategies for DMD in Italy;

- Research question 4 (economic impact): economic studies describing comparative economic evaluations (e.g. cost-effectiveness studies) or cost of illness analyses.

For all the research questions, only studies performed in Italy, or international multicenter studies with at least one Italian centre were included.

## 2.4. Literature screening process

The study selection process was performed independently by two review authors. Any disagreement was solved through discussion and, in the event of no agreement, a third reviewer was involved. Initially, the reviewers screened the records reading titles and abstracts, according to predefined inclusion criteria. Later, they evaluated the full-text of the potential eligible studies. The final studies included in the review have been described in the main text and in the tables, while a list of excluded studies along with the reasons for their exclusion is available as supplementary material (S3 Appendix). Bibliographic references were managed using the EndNote X7.4 software.

## 2.5. Data extraction

Data extraction was performed by one reviewer and verified by another reviewer using a standardized form. The following general information were extracted from the included studies: bibliographic data (first author, publication year and title), study characteristics (study design, study objective, location, study period, sample size), participant characteristics (type of disease, mean age), study outcomes (incidence/prevalence, quality of life scores, adherence measures, economic measures).

In addition, for each research question we extracted the following specific information:

- Research question 1 (epidemiology), secondary studies: number of included studies and their bibliographic references, epidemiological estimates (prevalence in the general population, birth prevalence, incidence). Primary studies: study population (N), epidemiological estimates.

- Research question 2 (quality of life): methods used for assessing QoL, subjects interviewed (patients and/or caregivers), presence of a reference population (e.g. healthy subjects), results.

- Research question 3 (adherence): definition of non-adherence, percentage of non-adherent patients, reasons for non-adherence, factors influencing (barriers / facilitators) the adherence.

- Research question 4 (economic impact): type of economic analysis, type of costs (direct, indirect, non-healthcare costs), discount rate, economic perspective, reference year, results.

## 2.6. Quality assessment of included studies

The quality assessment was performed by one reviewer and verified by another reviewer, using checklists specific to the study design of included studies. In particular, the following checklists were used: for systematic reviews, the JBI Critical Appraisal Checklist for Systematic Reviews and Research Synthesis [15]; for prevalence studies, the checklist by Hoy et al. [16]; for qualitative studies, the Critical Appraisal Skills Programme (CASP) checklist [17]; for economic studies, the Consensus Health Economic Criteria (CHEC) list [18].

## 2.7. Strategy for data synthesis

In our study protocol, we anticipated that a quantitative synthesis of the results through a meta-analysis would only be carried out if three or more included studies reported on the same outcome for any research question; moreover, to be combined, the studies should have to be similar in terms of PICO (patient/population, intervention, comparator, outcome). Assuming a high level of heterogeneity between the included studies, it was planned to perform random effects meta-analyses, providing cumulative estimates along with 95% confidence intervals. The heterogeneity between the included studies would have been assessed using the $I^2$ statistic, considering a statistical significance level of $p < 0.05$ for all analyses. The STATA 13/SE software was indicated for the statistical analyses. It was also specified in the protocol that, in all the cases where a meta-analysis was not considered feasible, the results would have been presented narratively.

## 3. Results

The literature searches in the electronic databases identified a total of 550 records. After removing duplicates, 373 records were screened through title/abstract and 35 of these were considered eligible for full-text evaluation. Twenty-two articles were excluded after full-text screening, while 13 articles were included in the final analysis. A list of excluded studies along with reasons for exclusion is provided in S3 Appendix. Among the 13 included studies, 3 studies were used for addressing the research question 1 (epidemiology), 8 studies for the research question 2 (quality of life), 1 study for the research question 3 (adherence), and 3 studies for the research question 4 (economic impact); two studies were used to address both the research questions 2 and 4. The literature selection process is shown in the Fig 1 (PRISMA Flow Diagram).

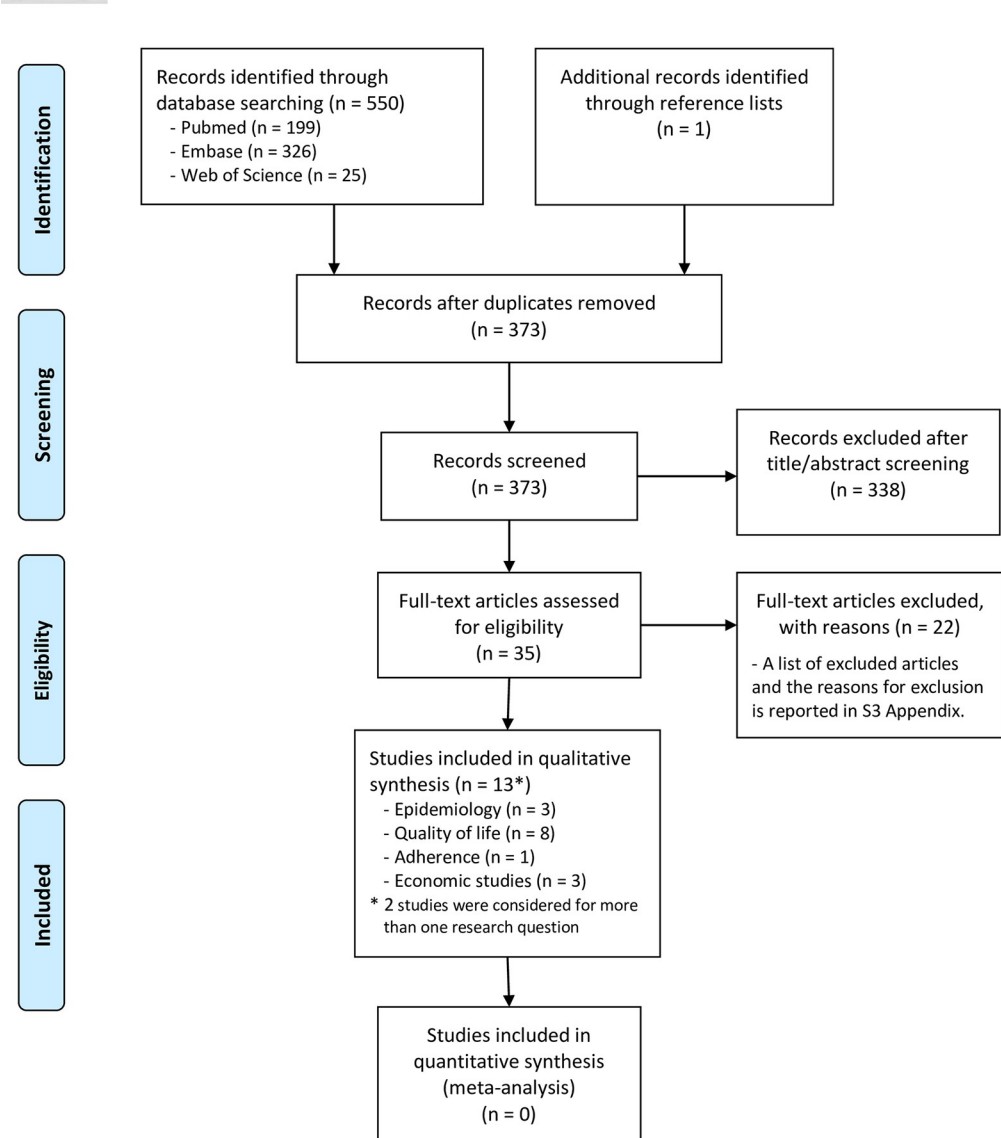

**PRISMA 2009 Flow Diagram**

Legend: *From:* Moher D, Liberati A, Tetzlaff J, Altman DG, The PRISMA Group (2009). *Preferred Reporting Items for Systematic Reviews and Meta-Analyses: The PRISMA Statement.* PLoS Med 6(7): e1000097. doi:10.1371/journal.pmed1000097. For more information, visit www.prisma-statement.org.

**Fig 1. PRISMA 2009 flow diagram.**

The results are presented narratively, without attempting a quantitative synthesis (meta-analysis) as the studies included in this review were heterogeneous in terms of study design, objectives and PICO.

In particular: for research question 1 (epidemiology), we found a recently published systematic review and meta-analysis that was judged to be of good quality, using the JBI Critical Appraisal Checklist for Systematic Reviews and Research Synthesis [15], and therefore we decided to present its results; for research question 2 (quality of life), the included studies differ

in terms of study design, objectives and type of tools used to measure quality of life; for research question 3 (adherence), we included only one study; for research question 4 (economic impact) there are no comparable economic measures in all three included studies. We reported the quality assessment for each research question in the S4 Appendix.

### 3.1. Epidemiology

We included 3 studies: a systematic review and meta-analysis [3], a systematic review [19], and an observational prevalence study [20]. The characteristics of the two reviews are described in the Table 1, while the observational study results are described in the main text.

The systematic review of Crisafulli et al. [3] reported a pooled global DMD prevalence from 22 studies of 7.1 cases (95% CI: 5.0–10.1) per 100,000 males and 2.8 cases (95% CI: 1.6–4.6) per 100,000 in the general population. Two out of 22 were Italian studies: Danieli et al. [21],

**Table 1. Characteristics of included secondary studies for epidemiology.**

| Study ID | Study design | Databases used and search dates | Included studies, N (Setting) | Disease | Prevalence in the general population | Birth prevalence |
|---|---|---|---|---|---|---|
| Crisafulli 2020 [3] | Systematic review and meta-analysis | MEDLINE and EMBASE, from databases inception to 1 October 2019 | Included in qualitative synthesis: n. 44. Included in meta-analysis: n. 40 | DMD | Worldwide: the pooled global prevalence (22 studies) is 7.1 cases (95% CI: 5.0–10.1; $I^2$ = 98.5%) per 100,000 males and 2.8 cases (95% CI: 1.6–4.6; $I^2$ = 87.4%) per 100,000 people (males + females). Italy: • Danieli 1977 [21]: Period prevalence per 100,000 males and females of all ages: 3.4 (2.8–4.2) per 100,000. • Siciliano 1999 [22]: Point prevalence per 100,000 males and females of any age: 1.7 (1.1–2.6) per 100,000. | Worldwide: the pooled global birth prevalence (29 studies) is 19.8 cases (95% CI: 16.6–23.6; $I^2$ = 89.8%) per 100,000 live male births. Italy: • Danieli 1977 [21]: 28.2 (22.1–35.8) per 100,000 live male births; • Danieli 1980 [23]: 28.2 (23.3–34.2) per 100,000 live male births; • Bertolotto 1981 [24]: 24.2 (19.3–30.5) per 100,000 live male births; • Nigro 1983 [25]: 21.7 (18.5–25.3) per 100,000 live male births; • Mostacciuolo 1987 [26]: 26.0 (34.4–53.9) per 100,000 live male births; • Merlini 1992 [27]: 25.8 (16.7–39.8) per 100,000 live male births. |
| Theadom 2014 [19] | Systematic review | Medline, CINAHL, Psychology and behavioral sciences collection, ProQuest, Scopus, and Web of Science, from 1 January 1960 to 30 October 2013 | Included in qualitative synthesis: n. 38 | Muscular dystrophies: DMD, BMD, congenital, myotonic, Emery-Dreifuss, facioscapulohumeral, oculopharyngeal, limb-girdle | Worldwide: • DMD (22 studies): range 1.0–7.7 per 100,000 people (general population); only studies with low risk of bias: 1.7–4.2 / 100,000. Italy: • DMD (5 studies: Danieli 1977 [21], Bertolotto 1981 [24], Mostacciuolo 1987 [26], Merlini 1992 [27], Siciliano 1999 [22]): range 1.7–3.4 per 100,000 people (general population) | Not reported. |

DMD, Duchenne muscular dystrophy; BMD, Becker muscular dystrophy.

through a retrospective chart review of patients hospitalized in 4 districts of Veneto region in the years 1952–1972, found a period prevalence of 3.4 (95% CI: 2.8–4.2) cases per 100,000 males and females of any age; Siciliano et al. [22], in a study conducted in Pisa (Tuscany) in the year 1997, reported a point prevalence of 1.7 (95% CI: 1.1–2.6) per 100,000 males and females of any age.

The review also reported a pooled global DMD birth prevalence from 29 studies of 19.8 cases (95% CI:16.6–23.6) per 100,000 live male births. Six Italian studies were included: the two studies by Danieli et al. [21, 23] (Veneto region, years 1952–1972) reported a period prevalence of 28.2 per 100,000 live male births; Bertolotto et al. [24] (Turin, years 1955–1974) a period prevalence of 24.2 per 100,000 live male births; Nigro et al. [25] (Campania region, years 1969–1980) a period prevalence of 21.7 per 100,000 live male births; Mostacciuolo et al. (1987) [26] (5 districts in the Veneto region, years 1955–1984) a period prevalence of 26.0 per 100,000 live male births; Merlini et al. [27] (Bologna, years 1970–1982) a period prevalence of 25.8 per 100,000 live male births.

All meta-analyses showed a substantial between-study heterogeneity (≥90%). Meta-regressions highlighted that the only covariate that reduced the heterogeneity of about 45% was the year in which the study began and its duration; older studies reported a higher DMD birth prevalence.

The methodological quality of the included studies was assessed through a modified version of the STrengthening the Reporting of OBservational studies in Epidemiology (STROBE) checklist; 36/44 (82%) studies were considered of moderate quality, 8/44 (18%) studies of low quality, while no study was of high quality.

The systematic review by Theadom et al. [19] describes the prevalence of all types of muscular dystrophies and includes 38 studies in the qualitative synthesis, 22 (58%) of which reporting the DMD prevalence. The included studies reported a prevalence range in the general population of 1.0–7.7 cases per 100,000. Considering only the studies with low risk of bias, the prevalence range narrows to 1.7–4.2 / 100,000.

This review included 5 Italian studies [21, 22, 24, 26, 27] that reported a DMD prevalence range of 1.7–3.4 per 100,000; all of these studies were also included in the Crisafulli review [3].

The quality assessment showed that 15/38 studies were considered having a low risk of bias, 14/38 an unclear risk of bias, and 9/38 a high risk of bias.

The last included study was the observational study of Mostacciuolo et al. (1993) [20], that updates a previous study above described [26] by using Western blot and DNA analyses as diagnostic criteria. The epidemiological estimates concern the incidence rate in the period 1959–1968 and the prevalence rate in the year 1980, in five districts of Veneto region. The reported DMD prevalence in the general population is 3.3 per 100,000, while the birth prevalence is 26.0 per 100,000 live male births.

## 3.2. Quality of life

To address the research question 2 (quality of life), we identified 8 studies. The main characteristics of included studies are reported in Table 2.

The study by Baiardini et al. [28] was carried out in an Italian centre located in Genova (Liguria region) and investigated the health-related quality of life and its possible determinants in 27 Duchenne muscular dystrophy children and in their parents. The tools used were the Children Health Questionnaire (CHQ)—Parent Form 50 and the Family Strain Questionnaire (FSQ); both questionnaires were completed by caregivers. Patients scored significantly lower than reference values of healthy subjects in 10 out of 15 CHQ dimensions. The use of a wheelchair (p <0.02) and pulmonary ventilator (p <0.001) are significantly associated with a lower health-related quality of life. Conversely, FSQ scores are not affected by patient characteristics.

**Table 2. Characteristics of included studies for quality of life.**

| Study ID | Population, mean (SD) age | Study design / period | Disease | Methods used for assessing QoL | Subjects interviewed | Control population | Results |
|---|---|---|---|---|---|---|---|
| Baiardini 2011 [28] | 27 patients 27 caregivers **Age** Patients: 11.3 (5.8) years Caregivers: 40.0 (7.8) years | Cross-sectional study (survey) Period: NR | DMD | Children Health Questionnaire (CHQ)—Parent Form 50; Family Strain Questionnaire (FSQ) | Caregivers | Healthy subjects (n = 788) (only CHQ questionnaire) | **CHQ—DMD Patient Score vs. healthy subjects (higher scores indicate a better health), mean (SD)** Global health: 43.3 (20.9) vs. 85.4 (16.1) Physical functioning: 16.5 (3.9) vs. 96.7 (11.9) Role/Social Limitations -Emotional: 52.7 (6.7) vs. 95.8 (15) Role/Social Limitations -Physical: 46.8 (7.4) vs. 95.3 (15.5) Bodily Pain and Discomfort: 63.7 (4.8) vs. 88.3 (17) General Health Perceptions: 37.8 (3.6) vs. 78.5 (13.9) Change in Health: 43.3 (4.3) vs. 59.7 (18.3) Emotional impact on parent: 42.5 (4.9) vs. 76.5 (24.2) Family activities: 71.6 (4.1) vs. 91.8 (13.1) Physical Summary score: 18.3 (2.0) vs. 54.5 (4.3) **FSQ, mean (SD) [possible range, higher values indicate problems]** Emotional burden: 7.7 (3.4) [0–14] Problem in social involvement: 2.7 (1.8) [0–7] Need of knowledge about the disease: 2.7 (1.1) [0–4] Satisfaction with family relationships: 3.2 (1.0) [0–4] Thoughts about death: 2.5 (1.3) [0–6] |
| Cavazza 2016 [29] | 87 patients 61 caregivers **Age** Patients: 13.5 (7.8) years Caregivers: 46.4 (9.1) years | Cross-sectional study (survey) Period: 2011–2013 | DMD | EQ-5D Barthel index Zarit Burden Interview | Patients and caregivers | General population (only EQ-5D questionnaire) | **EQ-5D (0 = death, 1 = perfect health), mean (SD)** Utilities adult patients: 0.19 (0.37) Utilities caregivers: 0.78 (0.27) **Visual Analogue Scale (VAS) (0 = worst health, 100 = best health), mean (SD)** VAS adult patients: 45.3 (15.9) VAS caregivers: 78.2 (16.2) **Barthel index (patients)** (0 = complete dependence, 100 = complete independence), mean (SD): 47.1 (31.6) **Zarit scale (caregivers)** (0 = lowest burden, 88 = highest burden), mean (SD): 24.5 (10.3) |
| Crescimanno 2019 [30] | 48 patients **Age**: 29.1 (7.1) years | Cross-sectional study (survey) Period: NR | DMD | Neuromuscular Quality of Life questionnaire (INQoL) Pittsburgh Sleep Quality Index (PSQI) Epworth Sleepiness Scale (ESS) Hospital Anxiety and Depression Scale (HADS) Composite Autonomic Symptom Score (Compass 31) | Patients | No control group | **INQoL (0 = lower symptom impact / best QoL, 100 = greater symptom impact / worst QoL), mean ± SD** Weakness: 61.6 ± 30.6 Locking 27.8 ± 20.5 Pain 43.9 ± 27.2 Fatigue: 32.9 ± 31.9 Activities: 65.1 ± 26.7 Independence: 81.1 ± 21.2 Body image: 41.2 ± 28.8 Relationships: 30.5 ± 29.1 Emotions: 31.9 ± 25.8 Global: 42.8 ± 19.2 **Assessment of psychological state: % subjects with abnormal scores** PSQI: 56.2% ESS: 0% HADS—Anxiety: 20.8% HADS—Depression: 10.4% Compass 31: 56.2% |
| Landfeldt 2014 [31] | 122 patients 122 caregivers **Age** Patients: 12 (8–17) Caregiver: 45 (41–50) | Cross-sectional study (survey) Period: July 2012-July 2013 | DMD | Health Utilities Index (patients) EQ-5D (caregivers) | Patients and caregivers | General population | **Health Utilities Index (0 = death, 1 = perfect health)** Patients: 0.52 (95% CI: 0.45–0.58) General population: the mean utility of a healthy 15-to19-year-old boy: 0.94 **EQ-5D** Caregivers: 0.84 (95% CI: 0.81–0.86) General population: the mean loss of caregiver quality of life in relation to the general population was 0.11 (considering the mean EQ-5D among the study participants of all countries) |

*(Continued)*

**Table 2.** (Continued)

| Study ID | Population, mean (SD) age | Study design / period | Disease | Methods used for assessing QoL | Subjects interviewed | Control population | Results |
|---|---|---|---|---|---|---|---|
| Magliano 2014 [32] | 336 parents of DMD (246) e BMD (90) patients **Age** • DMD patients: 10.0 (3.7) • BMD patients: 11.9 (3.6) • Parents of DMD patients: 41.2 (6.2) • Parents of BMD patients: 43.3 (6.6) | Cross-sectional study (survey) Period: 2012 | DMD, BMD | Family Problems Questionnaire (FPQ) and the Social Network Questionnaire (SNQ) | Caregivers | DMD vs BMD | **FPQ results** • Burden, mean (SD): DMD parents [1.8 (0.5)] vs BMD parents [1.4 (0.4), p < 0.05] • Feeling of loss (84.3% DMD vs 57.4% BMD) • Social stigma (44.2% DMD vs 5.5% BMD) • Need to awaken during the night (47.3% vs. 17.7%) • Neglect of hobbies (69.0% DMD vs 32.5% BMD) • Difficulties in work/household activities (55.5% vs. 18.9%) • Taking holidays (38.9% vs. 12.0%) • Financial difficulties (42.0% vs. 17.8%) • Belief that the disease has a negative influence on the psychological well-being (31.0% DMD parents vs 12.8% BMD parents) and on the social life of unaffected sibling (25.7% vs 18.4%). • Positive impact of caregiving experience on their lives: 66.0% of DMD parents vs 62.4% of BMD parents. |
| Magliano 2015 [33] | 502 patients (333 DMD, 129 BMD, 40 LGMD) and 502 parents Age Patients: 12.8 (5.6); Parents: 43.4 (7.4) | Cross-sectional study (survey) Period: 2012 | DMD, BMD, LGMD | Family Problems Questionnaire (FPQ) and the Social Network Questionnaire (SNQ) | Caregivers | DMD vs BMD vs LGMD | Considering all diseases, a total of 77.1% of relatives reported feelings of loss, 74.0% stated they cried or felt depressed and 59.1% neglected their hobbies. Burden was higher in relatives of DMD patients than in LGMD and BMD patients: • Practical burden, mean (SD): DMD 1.7 (0.6) vs LGMD 1.2 (0.4) vs BMD 1.4 (0.5), p <0.0001; • Psychological burden, mean (SD): DMD 2.0 (0.6) vs LGMD 1.9 (0.5), vs BMD 1.7 (0.5), (p <0.0001). |
| Messina 2016 [34] | 98 DMD patients **Age:** 8.4 (2.3) | Prospective cohort study Period: 1 year | DMD | The Paediatric Quality of Life Inventory (PedsQL), versions Child Self-Report and Parent Proxy-Report, with the following modules: Generic Core Scales (GCS); Multidimensional Fatigue Scale (MFS); Neuromuscular Module (NMM) | Patients and caregivers | No control group | At baseline, PedsQL items correlated with nearly all functional measures. Child Self-Report • PedsQL Generic Core Scales (GSC): Age (p < 0.001), NSAA (p = 0.01), 10 meters test (p = 0.009), Gowers test (p < 0.0001), 6MWT (p = 0.008). • PedsQL GSC– 1˚ domain (PFS): Age (p = 0.0004), NSAA (p = 0.002), 10 meters test (p = 0.0007), Gowers test (p < 0.0001), 6MWT (p = 0.02). The Parent Proxy-Report had similar results. On the Child Self-Report there was a significant decrease between baseline and 12 months: on the PedsQL GCS and the Generic Core Scales (from 74.5 to 71.5; p = 0.04) and its first domain (from 68.9 to 64.4; p = 0.03), in parallel with the decrease in functional outcome measures. The correlation between 12-month changes on PedsQL inventories and functional measures was negligible for almost all items. Similar results were obtained on the Parent Proxy-Report. |
| Orcesi 2014 [35] | 78 children with neuromuscular disorders and 81 healthy children **Age** Patients: 8.6 (2.3); Healthy children: 9.2 (2.6). | Development and administration of a questionnaire Period: NR | Neuromuscular disorders | Strips Of Life with Emoticons (SOLE) Questionnaire. | Patients and healthy children | Healthy children | The mean total SOLE score was significantly lower in children with neuromuscular disorders (141.9 ± 20.8) than in controls (163.7 ± 18.0) (t test: p <0.001). Even considering only DMD patients compared to the control group (43 DMD cases vs 44 controls), the mean SOLE total score was significantly lower in the DMD group (141.0 ± 20.5) than in the control group (159.7 ± 18.2; p <0.001). |

NR, not reported; DMD, Duchenne muscular dystrophy; BMD, Becker muscular dystrophy; LGMD, Limb-girdle muscular dystrophy; SD, standard deviation.

The multicenter study by Cavazza et al. [29] was aimed to determine the health-related quality of life of DMD patients and their caregivers in eight European countries. The Italian participants were 87 patients and 61 caregivers who filled out the EQ-5D, Barthel Index and Zarit Burden Interview questionnaires. As for the EQ-5D, the Italian adult patients obtained an average EQ-5D VAS score lower than the average score of the other countries (45.4 vs. 50.5) while for the caregivers the EQ-5D VAS score was higher than the international mean score (78.2 vs. 74.7). The average Barthel Index of the Italian population indicates severe dependence while the Zarit Burden Interview average score shows a mild-to-moderate burden. With reference to the entire study population, the authors conclude that the HRQoL of people with DMD is much lower than that of the general population and that patients with DMD rate their health status as more harm to their physical than mental health.

Crescimanno et al. [30] conducted a study in an Italian centre located in Palermo (Sicily) involving 48 long-term ventilated adult DMD patients, aimed to assess their QoL and possible determinants. The QoL was evaluated by the Neuromuscular Quality of Life questionnaire (INQoL); in addition, four other questionnaires were administered to assess their sleep quality, psychological status and symptoms of autonomic impairment. The overall INQoL score (42.8 ± 19.2) indicates a moderately altered quality of life, mainly due to high scores in the domains related to physical health (e.g. weakness, pain) and activities of daily living; on the other hand, the psychosocial domain appears to be much less affected by the disease. Loss of independence was the most important concern (mean score: 81.1 ± 21.2). Respiratory and muscular dysfunction and subjective sleep quality were severely impaired, and significantly correlated with several aspects of QoL; symptoms of sleepiness, psychological distress or autonomic dysfunction were marginally correlated to QoL.

The multicenter study by Landfeldt et al. (2014) [31] was carried out in Germany, Italy, United Kingdom (UK), and United States (USA), with the aim to estimate the cost of illness and economic burden of DMD. The QoL was assessed through the Health Utilities Index (for patients) and the EQ -5D (for caregivers). The patients' quality of life was similar across countries and significantly lower than the reference values of the general population. In particular, the Health Utilities Index for Italian patients was 0.52 (0.45–0.58), compared to a reference value of 0.94 for healthy boys aged 15–19. As for the Italian caregivers, the EQ-5D score obtained was 0.84 (0.81–0.86). On average, the loss of caregiver quality of life compared to the general population was 0.11 points (average score for patients of all countries).

The two papers by Magliano et al. [32, 33] described a cross-sectional study carried out in eight tertiary neuromuscular Italian centers from January to December 2012, that involved 502 families of patients with DMD, BMD, and LGMD. The first paper (2014) [32] was aimed to compare the burden in parents and healthy siblings of 4–17 year-old patients with DMD (n = 246) and BMD (n = 90). Parents filled out the Family Problems Questionnaire (FPQ) and the Social Network Questionnaire (SNQ) questionnaires. The mean burden of parents of DMD children was significantly higher [mean (SD): 1.8 (0.5)] compared to BMD [1.4 (0.4); p <0.05]. Parents of patients with DMD reported a higher burden with regard to the feeling of loss (84.3% DMD vs 57.4% BMD), social stigma (44.2% DMD vs 5.5% BMD) and neglect of hobbies (69.0% DMD vs 32.5% BMD). However, 66% of DMD parents and 62.4% of BMD parents reported that the caregiving experience had a positive impact on their lives. A minority of parents believed that the disease has a negative influence on the psychological well-being (31.0% DMD vs 12.8% BMD) and on the social life of healthy siblings (25.7% vs 18.4%).

The other paper by Magliano et al. (2015) [33] included 502 parents (333 DMD, 129 BMD, 40 LGMD) that completed the FPQ and SNQ questionnaires; in addition to the items considered in the previous study (2014) [32], this paper describes the answers to an additional open-ended question that asks key relatives what they suggest to improve care-givers' quality of life.

Most of relatives (77%) reported feelings of loss, 74% stated they cried or felt depressed and 59% neglected their hobbies. The burden was higher in relatives of patients with DMD than in patients with LGMD and BMD, respectively (burden of practical activities, mean [SD]: 1.7 [0.6] vs 1.2 [0.4] vs 1.4 [0.5], p <0.0001; psychological burden: 2.0 [0.6] vs. 1.9 [0.5], vs. 1.7 [0.5], p <0.0001). As for the answers to the open question, 43% of caregivers suggested improvements in the quality of care, 33% suggested improvements in welfare policies, 30% recommended providing psychological support to families and patients and 19% suggested greater investment in rare disease research. In addition, regression analyses showed a higher practical and psychological burden among relatives who were unemployed, who did not live with a partner, relatives of patients with higher level of functional disabilities, who spent more daily time in caregiving, having fewer social contacts, and who perceived that they had lower levels of support for the patient's emergencies from their social network.

The multicenter study by Messina et al. [34] involved 98 ambulant boys with DMD from 10 Italian centers, with the aim to assess if 12 month changes in function, assessed by 6MWT, NSAA, 10 meter timed walk/run and Gowers test, were associated to changes on HRQoL. At baseline, PedsQL items correlated with nearly all functional measures, i.e. worse functional measures corresponded to lower QoL levels. The Child Self-Report reports a significant decrease between baseline and 12 months on the PedsQL GCS (74.5 to 71.5; p = 0.04) and its first domain Physical Function Score (68.9 to 64.4; p = 0.03), corresponding to the decrease in functional outcome measures, indicating a decrease in QoL during the follow-up period. Conversely, the correlation between 12-month changes in PedsQL items and functional measures was not significant for almost all items. Similar results were obtained on the Parent Proxy-Report.

The paper of Orcesi et al. of 2014 [35] describes the development of The Strips Of Life with Emoticons Questionnaire (SOLE), a child-centred, multidimensional self-assessment tool for measuring quality of life in children with neuromuscular disorders. The questionnaire was tested in 78 children with neuromuscular disorders and 81 healthy children. Patients were recruited from 6 Italian tertiary centres, while controls (healthy children) were recruited from 4 nursery and primary schools in 2 areas of northern Italy (near Pavia and Turin). The neuromuscular disorders considered were: DMD n = 43; BMD n = 3; Duchenne-Becker muscular dystrophy n = 3; spinal muscular atrophy type II (SMA) n = 14; SMA type III n = 4; congenital myopathy n = 4; girdle dystrophy n = 3; congenital muscular dystrophy n = 1; myotonic dystrophy n = 1; myasthenia gravis n = 1; Emery-Dreifuss muscular dystrophy n = 1.

The mean total score of the SOLE questionnaire was significantly lower in children with neuromuscular disorders (141.9 ± 20.8) than in controls (163.7 ± 18.0) (p <0.001). Similar results were obtained by selecting only the DMD patients and comparing them with the males of the control group (43 DMD cases vs 44 controls): the mean total SOLE score was significantly lower in the DMD group (141.0 ± 20.5) than the control group (159.7 ± 18.2) (p <0.001).

## 3.3. Adherence

To address the research question 3 (adherence), we identified only the study by Landfeldt et al. of 2015 [11], aimed at comparing the experience reported by the patient (or his family) regarding the medical management of DMD in Germany, Italy, UK and USA, with the recommendations of clinical guidelines for the diagnosis and management of DMD, produced by an international, multidisciplinary group of experts [4, 36]. A total of 770 patient-caregiver pairs completed the questionnaire; of these, 122 couples (16%) were Italian. The questionnaire asked questions about the patient, his health status, visits to doctors and other healthcare

providers related to DMD, clinical tests and assessments, drug use, and access to medical aids and devices. The study highlighted a low adherence to the guidelines in all four countries. In particular, with reference to Italian couples, the percentage of patients undergoing visits by a general practitioner or paediatrician in the last six months was 34% (42/122) and this rate is the lowest observed (Germany 66%, UK 62%, USA 60%). Furthermore, despite the importance of routine follow-up due to the side effects associated with glucocorticoids, less than 27% (21/78) of Italian patients taking this class of drugs had visited a neuromuscular specialist in the last six months. Again, the Italian rate is the lowest observed among the countries (72% in Germany, 68% in UK, 62% in USA). Furthermore, it emerged that only 50% (25/50) of Italian patients with scoliosis had been visited by a physiotherapist in the last six months, compared to 80% in Germany, 55% in UK and 48% in USA. The rate of scoliosis patients visited by an orthopaedist in the last six months is also lower for Italian patients than in other countries: 18% (9/50) against 37% in Germany, 23% in UK, 29% in USA. Regarding the evaluation of lung function, also recommended every six months, the observed rate among Italian patients was 66% (37/56), compared to 81% in Germany, 61% in the UK and 62% in the USA. The observed rate among Italian ventilated patients was 46% (11/24), while it was 62% in Germany, 43% in the UK and 66% in USA. The study also reported that, despite the recommendations, the use of glucocorticoids was also highly variable among countries. In particular, Deflazacort was the most used drug in Italy (as well as in Germany and USA, whereas prednisolone was mostly prescribed in UK) according to an alternate-day dosing schedule. However, this scheme was not the most used in other countries where daily (USA) or intermittent (Germany and UK) administration prevails.

The access rate to orthotic devices and mobility aids also appeared variable: in particular, the Italian proportion was 65% for orthoses (Germany 29%, UK 65%, USA 70%) and 26% for wheelchairs; the latter is the lowest observed (Germany 55%, UK 76%, USA 58%).

## 3.4. Economic impact

To address the research question 4 (economic impact), we identified 3 studies (Table 3).

The study by Cavazza et al. [29] was aimed at determining the economic burden of DMD from a societal perspective, and measuring the health-related quality of life (HRQoL) of patients with DMD in Europe (see QoL paragraph). As for the Italian patients, the study reported average annual total costs per patient of € 41,547 (SD 35,811) in the year 2012; specifically, € 63,559 (SD 36,672) referring to adult patients and € 34,543 (SD 33,182) to paediatric patients. The annual direct healthcare costs per patient (e.g. drugs, exams, medical visits, etc.) amount to € 9,744 (SD 10,002), that is 23% of the total costs; in particular, € 20,601 (SD 11,230) if referring only to the adults and € 6,289 (SD 6,583) if referring only to children. Direct non-healthcare costs (e.g. professional and informal care) were much higher, being equal to 76% of the total costs (whole population: € 31,518, SD 32,408; adults: € 41,776, SD 33,691; children: € 28,258, SD 31,552). The labour productivity loss due to sick leave was on average €1,181 (SD 2,706) for adult patients, equal to approximately 2% of their total costs. The study also reports that, in all countries (except for Bulgaria), the average annual cost per patient with DMD is about ten times higher than the total per capita public health expenditure estimated by the World Health Organization.

The cross-sectional study by Landfeldt et al. of 2014 [31] estimated the total cost of illness and the economic burden of DMD from the perspective of the society and caregiver households. The study identified DMD patients through national registries in Germany, Italy, UK and USA, collecting questionnaires for a total of 770 patient-caregiver pairs; of these, 122 couples (16%) were Italian. As for Italy, the study reported an estimated total burden of illness for

**Table 3. Characteristics of included studies for economic impact.**

| Study ID | Population, mean (SD) age | Type of economic analysis | Perspective adopted (reference year; currency) | Direct healthcare annual costs per patient | Non-medical direct annual costs per patient | Indirect annual costs per patient | Other annual costs per patient |
|---|---|---|---|---|---|---|---|
| Cavazza 2016 [29] | 87 patients 61 caregivers **Age, mean (SD)** Patients: 13.5 (7.8) Caregivers: 46.4 (9.1) | Prevalence-based cost-of-illness analysis | Society (2012; EUR) | • Drugs: 321 (SD 224); adults only 249 (SD 222); children only 343 (SD 222) • Medical tests: 219 (SD 181); adults only 157 (SD 114); children only 238 (SD 194) • Medical visits: 1,863 (SD 2,329); adults only 1,358 (SD 2,133); children only 2,023 (SD 2,381) • Hospitalizations: 708 (SD 911); adults only 674 (SD: 935); children only 719 (SD 910) • Health material: 6,612 (SD 9,636); adults only 18,111 (SD 10,781); children only 2,953 (SD 5,546) • Healthcare transports: 23 (SD 132); adults only 53 (SD 217); children only 13 (SD 90) • Total: 9,744 (SD 10,002); adults only 20,601 (SD 11,230); only children 6,289 (SD 6,583) | • Professional carer: 75 (SD 703); adults only, no costs; children only 99 (SD 807) • Non-healthcare transports: 150 (SD 202); adults only 52 (SD 125); children only 181 (SD 213) • Social services: 655 (SD 1,902); adults only 1,865 (SD 3,272); children only 270 (SD 930) • Direct non-healthcare formal costs: 880 (SD 2,807); adults only 1,917 (SD 3,397); children only 550 (SD 1,950) • Main informal carer: 18,518 (SD 18,012); adults only 23,806 (SD 19,182); children only 16,835 (SD 17,439) • Other informal carer: 12,120 (SD 16,441); adults only 16,053 (SD 16,963); children only 10,869 (SD 16,202) • Direct non-healthcare informal costs: 30,638 (SD 34,453); adults only 39,859 (SD 36,144); children only 27,704 (SD 33,641) • Total: 31,518 (SD 32,408); adults only 41,776 (SD 33,691); children only 28,254 (SD 31,552) | • Sick leave: 285 (SD 2,659); adults only 1,181 (SD 5,412); children only, no costs • Labor productivity losses patients: 285 (SD 1,330); adults only 1,181 (SD 2,706); children only, no costs | - |
| Landfeldt 2014 [31] | 122 patients 122 caregivers **Age, median (IQR)** Patients: 12 (8–17) Caregivers: 45 (41–50) | Cost-of-illness analysis | Society and caregiver households (2012; Int$) | Mean (95% CI) • Drugs: 1,550 (890–4,650) • Tests and assessments: 600 (530–690) • Medical visits: 2,590 (1,970–3,440) • Hospitalizations: 1,420 (900–2,470) • Aids, devices and structural investments: 1,850 (970–4,450) | Mean (95% CI) • Social services (including home care and transports): 2,740 (1,640–5,380) • Informal care: 13,160 (11,270–15,280) **Total (includes direct healthcare and non-medical costs): 23,920 (20,420–28,300)** | Mean (95% CI) • Productivity loss: 18,220 (15,430–21,380) **Total (includes direct and indirect costs): 42,140 (36,940–47,730)** | Mean (95% CI) Intangible costs • QoL losses (e.g. pain, anxiety): 37,980 (32,400–43,550) **Total burden of illness (includes all costs): 80,120 (71,030–89,190) Household burden** • Out-of-pocket expenditure (insurance, copayments, structural investments): 7,550 (3,600–16,470) • Income loss: 620 (310–1,130) • Loss of leisure time: 12,440 (10,710–14,980) • Intangible costs (loss of QoL): 37,830 (30,220–41,760) **Total: 58,440 (50,200–68,900)** |

*(Continued)*

**Table 3.** (Continued)

| Study ID | Population, mean (SD) age | Type of economic analysis | Perspective adopted (reference year; currency) | Direct healthcare annual costs per patient | Non-medical direct annual costs per patient | Indirect annual costs per patient | Other annual costs per patient |
|---|---|---|---|---|---|---|---|
| Landfeldt 2017 [37] | NA | Economic model | Society (2012; EUR) | NR | NR | NR | Years of life lost in Italy due to DMD in 2012: 3,313 (3,297–3,329) Mean mortality cost in 2012: EUR 248 (247–250) million. |

SD, standard deviation; IQR, interquartile range; Int$, international dollar; QoL, quality of life; NA, not applicable; NR, not reported.

the society of Int$ (international dollar) 80,120 (71,030–89,190). The total annual cost per patient amounts to Int$ 42,140 (36,940–47,730) and include annual direct costs (medical and non-medical) per patient equal to Int$ 23,920 (20,420–28,300) (57% of the total cost), and indirect costs, due to losses of productivity, equal to Int$ 18,220 (15,430–21,380) (43%). In addition, intangible costs due to reduced quality of life (e.g. due to pain or anxiety) were estimated in Int$ 37,980 (32,400–43,550). The study also reports the annual family burden, due to out-of-pocket expenditures, income loss and reduced quality of life. This cost is estimated to be Int$ 58,440 (IC 50,200–68,900). In Italy, the annual direct cost per DMD patient was 8 times higher than the average public health expenditure per capita, while it was 10, 16 and 7 times higher than the average health expenditure per capita in Germany, UK and USA, respectively.

The study by Landfeldt et al. of 2017 [37] described an economic model for estimating the average annual cost of mortality caused by DMD in Germany, Italy, the UK and USA, according to a societal perspective. The life-years lost due to DMD in the reference year (2012) were estimated based on the incidence of the condition, which for Italy amount to 3,313 (IC 3,297–3,329) life-years, and subsequently the mortality cost was calculated using a willingness-to-pay (WTP) of € 75,000 per life-year gained. The national cost of DMD mortality in Italy was therefore estimated at € 248.5 (CI 247–250) million.

## 4. Discussion

This systematic review provides an up-to-date picture of the available evidence on DMD in Italy, focusing on four domains: epidemiology, quality of life, adherence to treatment and management strategies, and economic impact.

Three studies were identified to estimate the epidemiological burden of DMD in Italy, including a systematic review with meta-analysis, a systematic review and an observational study.

As for the prevalence in the general population, Italian studies reported a range of 1.7–3.4 cases per 100,000; this range is in line with the estimated worldwide prevalence of 2.8 per 100,000 (95% CI: 1.6–4.6) resulting from the Crisafulli meta-analysis [3]. Considering only males, the pooled global prevalence (no Italian studies) reported by Crisafulli [3] was 7.1 cases (95% CI: 5.0–10.1) per 100,000 males.

As for the birth prevalence, the Italian studies reported a range of 21.7–28.2 per 100,000 live male births; this range is slightly higher than the estimated worldwide birth prevalence of 19.8/100,000 live male births (95% CI: 16.6–23.6), but with overlapping confidence intervals [3].

All the included Italian studies were published between 1977 and 1999. As the dystrophin gene was discovered in 1986, prevalence studies prior to this date may have provided epidemiological estimates that may have been influenced by the lack of genetic diagnosis and

presumably overestimated the prevalence. Therefore, more recent epidemiological studies would be needed.

Regarding the quality of life of patients with DMD and their caregivers, we identified 8 studies published between 2011 and 2019. The studies were heterogeneous in terms of study objective, design, and tools used to measure the quality of life. The only tool to measure QoL used in more than one study was the EQ-5D, used by 2 studies [29, 31]. The Family Problems Questionnaire (FPQ) and the Social Network Questionnaire (SNQ) were used in the two 2 studies by Magliano [32, 33], but they described different sections of the questionnaires. All other studies used different tools.

Four studies [28, 29, 31, 35] compared DMD patients to healthy subjects/general population; all these studies showed a lower QoL in DMD patients and caregivers. Baiardini et al. [28] reported significantly lower scores for DMD children in 10 out of 15 CHQ dimensions. The highest score differences of DMD patients compared to the healthy population were observed in the dimensions "Physical functioning" (-80.25), "Role/Social limitations—physical" (-48.51), "Role/Social limitations—emotional" (-43.13), "Global health" (-42.07).

Cavazza et al. [29] reported utilities for adult patients of 0.19 (0.37) and utilities for caregivers of 0.78 (0.27); both scores are lower than those of the general population. Landfeldt et al. [31] reported utilities for DMD patients of 0.52 (0.45–0.58), compared to a reference value of 0.94 for healthy boys aged 15–19; the EQ-5D score obtained by caregivers was 0.84 (0.81–0.86) that was on average 0.11 points lower than the general population. Orcesi et al. [35] reported scores of patients with DMD significantly lower than the control population of healthy subjects using the SOLE questionnaire: 141.04 (+20.52) vs 159.69 (+18.24).

Two studies [32, 33] found a higher burden for the caregivers of DMD children compared to the caregivers of children with other neuromuscular disorders.

The most cited factors that negatively affect the quality of life of patients were motor and respiratory dysfunctions, weakness, pain, dependence in daily life activities and quality of sleep.

Among the critical issues highlighted by caregivers there were feelings of loss or sadness, social stigma and limitations in leisure activities. Despite these negative aspects, caregivers also expressed some positive aspects associated with assistance, considered as personal growth, development of a sense of resilience and altruism.

Furthermore, two studies compared the QoL of Italian patients and that of patients from other countries. One of them [29] reported that the EQ-5D VAS score of Italian adult patients (45.3) was lower than the average score of the countries participating in the study (50.5) and in particular, lower than that of Germany (62.5), UK (55.6), Spain (53.8) and Hungary (60.7), while it was higher than that of Bulgaria (39.6), France (30.0) and Sweden (28.5). On the other hand, as regards the EQ-5D VAS score of Italian caregivers (78.2), it was higher than the average of the countries (74.7) and respectively higher than that observed in all other countries, with the exception of the UK (81.5) and Sweden (80.0). The average Barthel Index of Italian patients (47.1) indicates severe dependence and was superior only to that of Spanish patients (38.9) (a lower score corresponds to a greater degree of patient dependence). Higher scores were found for Bulgaria (61.8), France (80.0), Germany (51.4) and Hungary (57.9); data for the UK and Sweden not comparable as they used a different scale. In contrast, Italian caregivers reported a subjective burden, measured by the Zarit Burden Interview, equal to 24.5, lower than that of Germany (34.7), the UK (31.3), Spain (32.5) and Sweden (37.3) but higher than the French one (14.5); data for Bulgaria and Hungary were not reported.

The second study [31] involved patients from Germany, Italy, the UK and USA. The quality of life of patients was measured using the Health Utilities Index, while that of caregivers using the EQ-5D tool. For both instruments, Italy showed slightly higher values (0 indicates death, 1

perfect health) than other countries. In Italian patients, the average utility derived from the Health Utilities Index was estimated at 0.52, compared to Germany (0.45), UK (0.43) and USA (0.45). Similarly, in Italian caregivers the EQ-5D utility estimate was 0.84, compared to Germany (0.79), UK (0.82) and USA (0.81).

Regarding adherence, the only included study [11] did not report data on adherence to a specific treatment, but rather described the compliance of real-world DMD care to clinical guidelines recommendations in four countries having different healthcare systems: Italy, Germany, UK and USA. Italian patients showed the lowest compliance compared to other countries for neuromuscular, and cardiac specialist visits, physiotherapy, and access to medical devices and aids (e.g. wheelchairs).

As for the economic impact of DMD in Italy, we found two cost of illness studies [29, 31] and an economic model [37] estimating the DMD mortality costs.

However, the data collected were not directly comparable as they were expressed in different currencies. Estimates were then converted into 2021 Euros: the annual per patient direct healthcare costs ranged from € 5,142 to € 10,641 and direct non-healthcare costs from € 14,825 to € 34,419; indirect costs (productivity loss) were not comparable because in one study [29] they referred to adult patients, while in another study [31] to caregivers. The impact of the intangible costs related to QoL and the economic burden on household were estimated at € 34,681 and € 48,782 per patient/year, respectively. Landfeldt (2017) et al. [37] reported a mortality DMD cost in Italy of € 271 million, compared to € 292 million in UK, € 366 million in Germany, and € 1,319 million in USA.

In addition to the three included economic studies, we found the conference abstract by Fabriani et al. (2014) [38] aimed to estimate the average annual direct and indirect costs (currency: EUR 2012) of DMD in Italy through an economic model, that was excluded from our main analysis due to insufficient information reported. This study estimated an annual direct healthcare cost for the Italian NHS of € 7,475,596 (EUR 2021: € 8,163,573), an indirect cost of € 474,634,836 (EUR 2021: € 518,315,323), and a nonmedical cost of € 12,944,879 (EUR 2021: € 14,136,194). Furthermore, as for private expenditure, the study estimated a direct cost of € 2,910,506 (EUR 2021 € 3,178,359), and a nonmedical cost of € 185,333,744 (EUR 2021: € 202,389,947).

## 4.1. Strengths and limitations

This review is based on a study protocol registered in the PROSPERO database; the reporting of the review follows the PRISMA guidelines; a comprehensive literature search was performed in three large electronic databases; the literature selection process, data extraction and quality assessment were performed by two independent reviewers.

Potential limitations concern the time horizon considered for the literature search (2010–2023) and that we considered only published studies, without searching the grey literature; these limitations could have reduced the sensitivity of the search.

## 5. Conclusions

This systematic review provides an updated description of the DMD burden in Italy: however, the body of evidence on DMD from Italian data is limited. Italian epidemiological studies date back to the 80s and 90s. Regarding adherence, a study found poor compliance of Italian patients with the recommendations of clinical guidelines. As for the economic impact, two studies reported an annual cost of illness for DMD in Italy converted into 2021 Euros of € 35,000–46,000 per capita, while adding also the intangible cost related to QoL the total cost amounted to € 70,000; a study reported a mortality cost in Italy of € 271 million. The average

annual cost per DMD patient was 8–10 times higher than the average Italian public health expenditure per capita. The evidence base on quality of life was wider, although highly heterogeneous, and has shown a significant burden of DMD for both patients and caregivers compared to healthy subjects and also compared to patients with other neuromuscular diseases.

## Supporting information

**S1 Appendix. PRISMA-2009 reporting checklist.**
(DOC)

**S2 Appendix. Literature search strategies.**
(DOCX)

**S3 Appendix. Excluded studies.**
(DOCX)

**S4 Appendix. Quality assessment.**
(DOCX)

## Author Contributions

**Conceptualization:** Massimiliano Orso, Antonio Migliore, Barbara Polistena, Daniela d'Angela, Federico Spandonaro, Marika Pane.

**Data curation:** Massimiliano Orso, Antonio Migliore.

**Formal analysis:** Massimiliano Orso, Antonio Migliore, Barbara Polistena, Marika Pane.

**Investigation:** Antonio Migliore, Barbara Polistena.

**Methodology:** Massimiliano Orso, Antonio Migliore, Barbara Polistena, Marika Pane.

**Project administration:** Barbara Polistena, Daniela d'Angela, Federico Spandonaro.

**Supervision:** Barbara Polistena, Federico Spandonaro.

**Writing – original draft:** Massimiliano Orso, Antonio Migliore, Barbara Polistena, Eleonora Russo, Francesca Gatto, Mauro Monterubbianesi, Daniela d'Angela, Federico Spandonaro, Marika Pane.

**Writing – review & editing:** Massimiliano Orso, Antonio Migliore, Barbara Polistena, Eleonora Russo, Francesca Gatto, Mauro Monterubbianesi, Daniela d'Angela, Federico Spandonaro, Marika Pane.

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
