## [Decision Letter · Decision Letter 0]

21 Mar 2023

PONE-D-23-04901Duchenne muscular dystrophy in Italy: a systematic review of epidemiology, quality of life, treatment adherence, and economic impactPLOS ONE

Dear Dr. Massimiliano Orso,

Thank you for submitting your manuscript to PLOS ONE. After careful consideration, we feel that it has merit but does not fully meet PLOS ONE’s publication criteria as it currently stands. Therefore, we invite you to submit a revised version of the manuscript that addresses the points raised during the review process.

We look forward to receiving your revised manuscript.

Kind regards,

Omar Enzo Santangelo

Academic Editor

PLOS ONE

Journal Requirements:

Reviewers' comments:

Reviewer's Responses to Questions

**Comments to the Author**

1. Is the manuscript technically sound, and do the data support the conclusions?

Reviewer #1: Yes

Reviewer #2: Yes

2. Has the statistical analysis been performed appropriately and rigorously? 

Reviewer #1: N/A

Reviewer #2: N/A

3. Have the authors made all data underlying the findings in their manuscript fully available?

Reviewer #1: Yes

Reviewer #2: Yes

4. Is the manuscript presented in an intelligible fashion and written in standard English?

Reviewer #1: Yes

Reviewer #2: No

5. Review Comments to the Author

Reviewer #1: Dear authors,

This paper was well-written and interesting. I think it will be a useful addition to the literature. The authors could not do a meta-analysis, but this is due to lack of literature. I do think the paper is too long. I see a potential to split it into two papers because there is sufficient material.

Kindly find my suggestions below:

Introduction

- In the introduction, the authors should clarify what their paper is expecting to add to the review and meta-analysis on the topic published by Crisafulli et al. I can see that this paper adds to the literature but it should be clearer for someone who doesn't know the previous literature. To me, the major contribution is the section on quality of life.

Method

- Please replace "searches" with "a literature search" or similar throughout the paper.

- Please be more specific when you refer to adherence. Adherence to what? Pharmacotherapy? Physiotherapy? Occupational therapy? Arguably all are important for people with DMD.

- Overall the methods seem very well-conducted. Well done.

Results

- I suggest the authors exercise caution in making a statement such as "In particular: for research question 1 (epidemiology), we found a systematic review with meta-analysis which was recently published and had good quality". If the authors assessed the quality of studies using GRADE or Newcastle-Ottawa/similar scales, these should be mentioned in the methods.

- Suggest being more specific on what the papers you identified about adherence concerned, as above. It is also necessary to be more specific when writing about adherence. For example, the authors write "The study highlighted a low adherence to the guidelines in all four countries." Which guidelines? Please be more specific.

Reviewer #2: This manuscript is a systematic review of studies addressing the epidemiology, quality of life, treatment adherence, and economic impact of Duchenne muscular dystrophy (DMD). While the authors do a reasonable job in reviewing the literature in the four broad areas, the manuscript is challenging to read. It is verbose, repetitive, and fails to summarize the information in a satisfactory manner. The epidemiology portion seems problematic because the prevalence rates are estimated by different papers in different time windows. It is difficult to assess whether pooling these estimates makes sense.

It might be helpful to have a professional editor rewrite the paper so that it is more effective at summarizing the retained studies. It might also be helpful to create a figure for each broad area to summarize the findings more visually.

6. PLOS authors have the option to publish the peer review history of their article (what does this mean?). If published, this will include your full peer review and any attached files.

Reviewer #1: No

Reviewer #2: No

---

## [Author Response · Author response to Decision Letter 0]

8 Jun 2023

RESPONSE: We deleted the sentence “data not shown” from the Table 2 because it was referred to the fact that in the study by Cavazza et al. EQ-5D reference values for the general population have not been reported. Authors only wrote: “Therefore, the HRQOL of people with DMD is much lower than that of the general population”, without reporting the actual EQ-5D scores for the general population.

Reviewers' comments:

Reviewer's Responses to Questions

Comments to the Author

1. Is the manuscript technically sound, and do the data support the conclusions?

Reviewer #1: Yes

Reviewer #2: Yes

2. Has the statistical analysis been performed appropriately and rigorously?

Reviewer #1: N/A

Reviewer #2: N/A

3. Have the authors made all data underlying the findings in their manuscript fully available?

Reviewer #1: Yes

Reviewer #2: Yes

4. Is the manuscript presented in an intelligible fashion and written in standard English?

Reviewer #1: Yes

Reviewer #2: No

5. Review Comments to the Author

Reviewer #1: Dear authors,

This paper was well-written and interesting. I think it will be a useful addition to the literature. The authors could not do a meta-analysis, but this is due to lack of literature. I do think the paper is too long. I see a potential to split it into two papers because there is sufficient material.

RESPONSE: Thank you for your positive comment. We agree that our paper is quite long and, in order to shorten the text, we decided to propose the quality assessment sections as supplementary material. This solution was accepted by the Editor. The paper is now substantially shorter and more readable.

We discussed with our co-authors the opportunity to split the paper in two and agreed to not do that as we prefer to publish a single paper in order to provide a comprehensive picture of DMD in Italy, describing the four areas together. We followed a similar approach in another review published on Plos One on paediatric growth hormone deficiency (GHD) in Italy (https://journals.plos.org/plosone/article?id=10.1371/journal.pone.0264403).

Kindly find my suggestions below:

Introduction

- In the introduction, the authors should clarify what their paper is expecting to add to the review and meta-analysis on the topic published by Crisafulli et al. I can see that this paper adds to the literature but it should be clearer for someone who doesn't know the previous literature. To me, the major contribution is the section on quality of life.

RESPONSE: Our systematic review was aimed to investigate the following four research questions: 1) the epidemiology of DMD in Italy; 2) the quality of life of patients with DMD and their caregivers in Italy; 3) the adherence to the different management strategies for DMD in Italy and to identify factors that could facilitate or hamper the adherence; 4) the economic impact of DMD in Italy.

As for the first research question (epidemiology), we included in our review both secondary studies (systematic or narrative reviews) and primary studies (cross-sectional, cohort) describing prevalence and/or incidence of DMD in Italy. Our aim was to summarize and update the existing evidence about epidemiology. Owing that among our included studies we found the recent systematic review and meta-analysis by Crisafulli et al., we described their results and in addition we described the results of another systematic review and those of an additional epidemiological primary study that met our inclusion criteria.

In conclusion, to address your comment, our review presents an evidence update, considering that Crisafulli et al. performed their searches in October 2019 and our search was performed in January 2023, and in addition includes updated evidence on the other research questions, particularly on the quality of life of DMD patients and their caregivers, as you pointed out. We slightly amended the text in different parts to reflect this.

Method

- Please replace "searches" with "a literature search" or similar throughout the paper.

RESPONSE: we amended the text as suggested.

- Please be more specific when you refer to adherence. Adherence to what? Pharmacotherapy? Physiotherapy? Occupational therapy? Arguably all are important for people with DMD.

RESPONSE: We were interested in assessing the adherence to all the management strategies for DMD, including those you mentioned. We have changed the text as follows: “3) the adherence to the different management strategies for DMD in Italy, such as pharmacotherapy, physiotherapy and occupational therapy, and to identify factors that could facilitate or hamper the adherence”.

- Overall the methods seem very well-conducted. Well done.

RESPONSE: Thank you for your positive comment.

Results

- I suggest the authors exercise caution in making a statement such as "In particular: for research question 1 (epidemiology), we found a systematic review with meta-analysis which was recently published and had good quality". If the authors assessed the quality of studies using GRADE or Newcastle-Ottawa/similar scales, these should be mentioned in the methods.

RESPONSE: We assessed the methodological quality of the systematic review authored by Crisafulli et al. using the JBI checklist, as explained later in the text (quality assessment sections; now included as supplementary material). We have included the following sentence in paragraph 3: “In particular: for research question 1 (epidemiology), we found a recently published systematic review and meta-analysis that was judged to be of good quality, using the JBI Critical Appraisal Checklist for Systematic Reviews and Research Synthesis [15], and therefore we decided to present its results”.

- Suggest being more specific on what the papers you identified about adherence concerned, as above. It is also necessary to be more specific when writing about adherence. For example, the authors write "The study highlighted a low adherence to the guidelines in all four countries." Which guidelines? Please be more specific.

RESPONSE: We added some information in the text, also citing the guidelines: “To address the research question 3 (adherence), we identified only the study by Landfeldt et al. of 2015 [11], aimed at comparing the experience reported by the patient (or his family) regarding the medical management of DMD in Germany, Italy, UK and USA, with the recommendations of clinical guidelines for the diagnosis and management of DMD, produced by an international, multidisciplinary group of experts [4, 36]. A total of 770 patient-caregiver pairs completed the questionnaire; of these, 122 couples (16%) were Italian. The questionnaire asked questions about the patient, his health status, visits to doctors and other healthcare providers related to DMD, clinical tests and assessments, drug use, and access to medical aids and devices”.

Reviewer #2: This manuscript is a systematic review of studies addressing the epidemiology, quality of life, treatment adherence, and economic impact of Duchenne muscular dystrophy (DMD). While the authors do a reasonable job in reviewing the literature in the four broad areas, the manuscript is challenging to read. It is verbose, repetitive, and fails to summarize the information in a satisfactory manner. 

RESPONSE: We agree that our paper is relatively long. However we think that this is due to the fact that we addressed four different research questions. In addition, we reported the results narratively, describing each study separately, and this had contributed to the manuscript length. A summary of results is reported in the Discussion section.

In order to shorten the text, in agreement with the Editor, we presented the quality assessment sections as supplementary material. In this way, the length of the paper has been significantly reduced.

The epidemiology portion seems problematic because the prevalence rates are estimated by different papers in different time windows. It is difficult to assess whether pooling these estimates makes sense.

RESPONSE: As for epidemiology, we included three studies: a systematic review and meta-analysis (Crisafulli et al.), another systematic review (Theadom et al.) and an observational epidemiological study (Mostacciuolo et al.). Crisafulli et al. reported pooled global DMD prevalence from 22 studies published between 1982 and 2016; they also reported pooled DMD birth prevalence from 29 studies published between 1977 and 2019. Heterogeneity was high in both meta-analyses (87% and 90%, respectively). Meta-regression was performed to investigate possible source of heterogeneity. The covariates selected were: the continent where the studies were carried out; the year the study began and its duration; the study design. As for the birth prevalence meta-analyses, the only covariate that significantly reduced the between-study heterogeneity (of about 45%) was the study period. In fact, older studies reported higher birth prevalence.

We added the following sentence in the text: “All meta-analyses showed a substantial between-study heterogeneity (≥90%). Meta-regressions highlighted that the only covariate that reduced the heterogeneity of about 45% was the year in which the study began and its duration; older studies reported a higher DMD birth prevalence”.

It might be helpful to have a professional editor rewrite the paper so that it is more effective at summarizing the retained studies. It might also be helpful to create a figure for each broad area to summarize the findings more visually.

RESPONSE: As explained in our previous response to your first comment, in agreement with the Editor, we presented the quality assessment sections as supplementary material. In this way, the length of the paper has been significantly reduced.

As for your suggestion to create a figure for each research question, we do not think that this is feasible; we had presented the PRISMA flow diagram showing the literature search process and we believe this help the readers in visualize the overall process (see Figure 1).

---

## [Decision Letter · Decision Letter 1]

13 Jun 2023

Duchenne muscular dystrophy in Italy: a systematic review of epidemiology, quality of life, treatment adherence, and economic impact

PONE-D-23-04901R1

Dear Dr. Orso,

We’re pleased to inform you that your manuscript has been judged scientifically suitable for publication and will be formally accepted for publication once it meets all outstanding technical requirements.

Kind regards,

Omar Enzo Santangelo

Academic Editor

PLOS ONE

Additional Editor Comments (optional):

Reviewers' comments:

Reviewer's Responses to Questions

**Comments to the Author**

1. If the authors have adequately addressed your comments raised in a previous round of review and you feel that this manuscript is now acceptable for publication, you may indicate that here to bypass the “Comments to the Author” section, enter your conflict of interest statement in the “Confidential to Editor” section, and submit your "Accept" recommendation.

Reviewer #1: All comments have been addressed

Reviewer #2: All comments have been addressed

2. Is the manuscript technically sound, and do the data support the conclusions?

Reviewer #1: Yes

Reviewer #2: Yes

3. Has the statistical analysis been performed appropriately and rigorously? 

Reviewer #1: Yes

Reviewer #2: N/A

4. Have the authors made all data underlying the findings in their manuscript fully available?

Reviewer #1: Yes

Reviewer #2: Yes

5. Is the manuscript presented in an intelligible fashion and written in standard English?

Reviewer #1: Yes

Reviewer #2: Yes

6. Review Comments to the Author

Reviewer #1: The authors have addressed my comments adequately. I have nothing more to add. I think this paper will be a useful addition to the literature.

Reviewer #2: The paper remains long but it is an ambitious effort that will likely provide useful and important data to the DMD community.

7. PLOS authors have the option to publish the peer review history of their article (what does this mean?). If published, this will include your full peer review and any attached files.

Reviewer #1: **Yes: **Janet Sultana

Reviewer #2: No

---

## [Editor Report · Acceptance letter]

19 Jun 2023

PONE-D-23-04901R1 

Duchenne muscular dystrophy in Italy: a systematic review of epidemiology, quality of life, treatment adherence, and economic impact 

Dear Dr. Orso:

I'm pleased to inform you that your manuscript has been deemed suitable for publication in PLOS ONE. Congratulations! Your manuscript is now with our production department. 

Kind regards, 

on behalf of

Dr. Omar Enzo Santangelo 

Academic Editor

PLOS ONE